# Marangoni Patterns in a Non-Isothermal Liquid with Deformable Interface Covered by Insoluble Surfactant

**Alexander B. Mikishev [1],\* and Alexander A. Nepomnyashchy [2]**

[1]  Department of Engineering Technology, Sam Houston State University, Huntsville, TX 77341, USA

[2]  Department of Mathematics, Technion-Israel Institute of Technology, Haifa 32000, Israel

\*   Correspondence: amik@shsu.edu

**Abstract:** Marangoni patterns are created by instabilities caused by thermocapillary and solutocapillary stresses on the deformable free surface of a thin liquid layer. In the present paper, we consider the influence of the insoluble surfactant on the selection and modulational instability of stationary Marangoni patterns near their onset threshold. The basic governing parameters of the problem are the Biot number characterizing the heat-transfer resistances of and at the surface, the Galileo number indicating the role of gravity via viscous forces, and the elasticity number specifying the influence of insoluble surfactant on the interfacial dynamics of the liquid. The paper includes a review of the previous results obtained in that problem as well as new ones.

**Keywords:** Marangoni convection; instability; surfactant

## 1. Introduction

Interfacial phenomena are significant in many natural and technological processes. A basic phenomenon is the interfacial pattern formation discovered by H. Bénard at the end of the 19th century [1]: under homogeneous external conditions, the system becomes spatially inhomogeneous. The correct theoretical explanation of these experimental results was proposed more than 50 years later in the 1950s of the 20th century by Block [2] and, independently, by Pearson [3]. They both explained the interfacial pattern formation as a result of surface tension variations of a liquid free surface due to temperature changes along the surface.

The interest in the pattern formation was awoken by the seminal paper of Turing [4], who predicted theoretically a spontaneous pattern generation in a chemical system with two reacting and diffusing chemicals (found experimentally in the 1990s [5]), and by the paper of Normand, Pomeau, and Velarde [6], who attracted the attention of physicists to the convective pattern formation. The most typical convective patterns are rolls, squares, and hexagons. Numerous examples of patterns can be found in the recent publications [7,8]. Now, we know that the process of pattern formation is typical for physical, chemical, and biological systems which are far from the thermodynamic equilibrium state.

The longwave Marangoni convection is the focus of the special interest of the experts on pattern formation because of a rich gallery of patterns that can be formed. There exist two kinds of longwave Marangoni instabilities: (i) the first is the already mentioned Pearson's mode which is possible in the case of poorly conducting boundaries, and (ii) the mode described by Scriven and Sternling [9] existing in the case of a thin film. The evolution of longwave instability is described by a slowly evolving "active" variable that determines the nonlinear dynamics. The study of the pattern formation caused by the Pearson's mode was started in the works of Sivashinsky [10] and Knobloch [11]. In this case, the temperature plays the role of the active variable. In the case of the mode of Scriven and Sternling, the active variable is the free-surface deformation. The nonlinear evolution of this mode was described by Davis [12].

Recently, a remarkable interest in the Marangoni patterns generated by the longwave Marangoni instability was motivated by the work of Shklyaev et al. [13]. The authors

showed that in the interval of wavenumbers $k \sim O(Bi^{1/2})$ ($Bi \ll 1$ is the Biot number, defined as the ratio of heat-transfer resistances inside of and at the surface of the liquid), the temperature disturbance and the surface deformation are both active variables. Their interaction can generate large-scale monotonic and oscillatory instabilities, which produce stationary patterns and wave patterns [14].

In applications such as inject printing [15,16], spray cooling [17], the fabrication of DNA/RNA microarrays [18], photolithography [19], and various pulmonary therapies [20], one has to deal with the influence of surfactant contaminants. The nonuniform deposition of surfactant over the liquid surface can be a cause of special surfactant-induced solutal Marangoni stresses that significantly change the patterning profile of a liquid. Even a small concentration of impurities can cause the solutal stresses that interact with the thermal Marangoni flow [21,22]. Recently, the experimental and theoretical works in colloid science showed that the formation of unusual patterns (from rings and the "Marangoni ridge" to hexagonal arrays) are related to the Marangoni instabilities of deposited liquid films [23–26]. A highly ordered pattern is observed when a milimeter-size drop of dichloromethane spreads on an aqueous substrate under a change in Marangoni stresses [27]. The surfactant adsorption at a droplet interface is capable of modulating the flow patterns via the wetting shape, varying the liquid–vapor interfacial properties, and inducing superficial flows [28].

The conservation of the amount of the absorbed insoluble surfactant is the origin of an additional slow long-scale process: the redistribution of the adsorbed surfactant due to the surface flow and diffusion. The surfactant concentration is one more active variable. Though the presence of an insoluble surfactant on the surface provides no specific instability mechanism, it can significantly modify the development of instability generated by the thermocapillary instability.

In the present work, we review the latest results on the large-scale Marangoni convection in a liquid layer with an insoluble surfactant spread over a deformable liquid interface. The literature on the Marangoni convection with a soluble surfactant was reviewed in detail by Shklyaev and Nepomnyashchy in [8]. We present a weakly nonlinear analysis of the patterns on square and rhombic lattices in the Fourier space near the instability threshold. The modulation of these convective patterns is considered using the Newell, Whitehead, and Segel [29,30] approach, describing the interaction of the disturbances with various wavenumbers close to the critical one. As a result, this modulation is described by the Ginzburg–Landau-type equation for the amplitude of patterns coupled with equations for surface distortion and surfactant concentration. We discuss the role of 2D modulation on the roll stability. Several cases have been revealed. In each case, a specific stability map in terms of the Biot and Galileo numbers is constructed. In all these cases, the special role of insoluble surfactant is investigated. The results of the 1D modulation of rolls have been formerly published in [31]. All other results are presented here for the first time. At the end, the perspectives of the future research are discussed.

## 2. Description of Longwave Marangoni Convection with Insoluble Surfactant

### 2.1. Formulation of the Problem

The presence of an insoluble surfactant changes the interfacial stresses on the deformable surface of the thin liquid layer. In the case of a non-isothermal liquid layer (the liquid is subjected to a transverse fixed temperature gradient $-a$, ($a > 0$) [32]), the surfactant provides an additional factor, a solutocapillary mechanism, to generate Marangoni stresses on the liquid surface. The surface tension can be written as a linear function of both the temperature $T$ and concentration of surfactant $\Gamma$,

$$\sigma = \sigma_0 - \sigma_1(T - T_0) - \sigma_2(\Gamma - \Gamma_0),$$

and here, $\sigma_0$ is the reference value of the surface tension $\sigma_1 = -\partial_T \sigma$, $\sigma_2 = -\partial_\Gamma \sigma$, and $T_0$ and $\Gamma_0$ are the reference values of the corresponding variables at the surface in the absence of convection. The absorbed surfactant at the free surface is convected and diffuses over the free-liquid interface but not into the bulk. The parameter that describes the relation

between surfactant diffusivity $D_0$ and thermal diffusivity $\chi$ of the liquid is called the Lewis number, $L = D_0/\chi$. Typically, the Lewis number is small, $L \leqslant 0.01$.

The parameter characterizing the influence of the insoluble surfactant on the interfacial dynamics of the liquid is the elasticity number, $N = \sigma_2 d_0 \Gamma_0/\eta\chi$. It determines the significance of the solutocapillary effect as compared to viscous forces. It is a changeable parameter. Here, $d_0$ is a mean thickness of the liquid layer and $\eta$ is a dynamic viscosity ($\eta = \rho\nu$, $\rho$ is a density of the liquid, and $\nu$ is a kinematic viscosity of the liquid).

Let us discuss the characteristic values of the surfactant parameters, $N$ and $L$. While the elasticity number, $N$, which depends on the surfactant concentration, can vary in a relatively large interval, the value of the surfactant Lewis number has a significant limitation. As an example of an insoluble surfactant, let us consider $C_{12}EO_8$, produced by Nikko Chemical Co. (Tokyo, Japan); for its properties, see [33]. The typical surfactant values of the surface surfactant diffusion coefficient are $(3.0\text{–}4.0)\times10^{-10}$ (m$^2$/s), [34]. The surfactant is located at the air/water interface, and for other parameters, we take the physical values of the water at the room temperature. It means the value of the thermal diffusivity coefficient, $\chi$, is $1.3 \times 10^{-7}$ (m$^2$/s), and the kinematic viscosity, $\nu$, is $0.9 \times 10^{-6}$ (m$^2$/s). For the calculation, we fix the Lewis number at $L = 0.003$.

The schematic of the problem is presented in Figure 1. Here, the liquid layer is bounded at the bottom by a rigid substrate and has an upper free surface. The coordinate axes $(x, y)$ are in the plane of the substrate and the $z$ axis points upward.

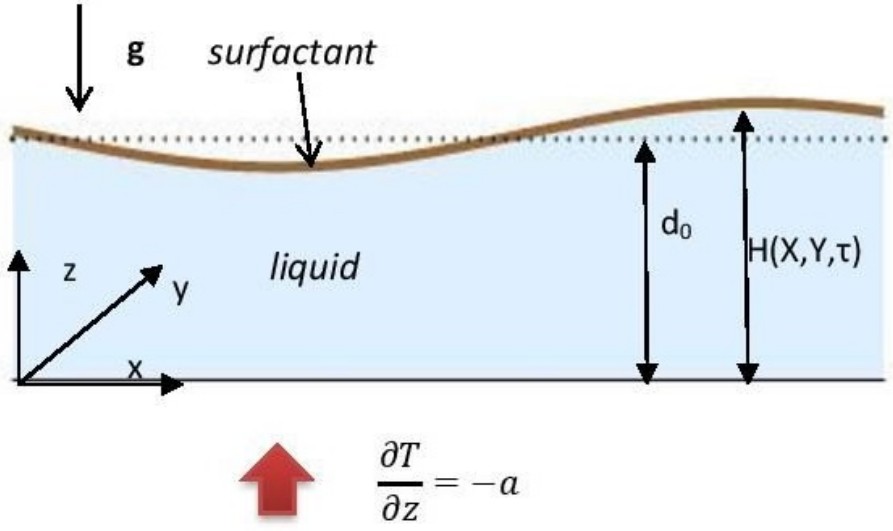

**Figure 1.** Schematic of the problem.

The typical additional dimensionless parameters of the instability problems are the Marangoni number, $M = \sigma_1 a d_0^2/\eta\chi$, the ratio of the thermocapillary surface force to the viscous force; the Galileo number, $G = g d_0^3/\nu\chi$, the ratio between the gravity force and the viscous force; the inverse capillary number, $\Sigma = \sigma_0 d_0/\eta\chi$, the ratio between the surface tension force and the viscous force; and the Biot number, $Bi = q d_0/\Lambda_T$ ($q$ is the heat-transfer coefficient and $\Lambda_T$ is the thermal conductivity), the ratio of the heat-transfer resistances inside of and at the surface of the liquid.

The mathematical description of the interfacial dynamics includes the surface transport equation for the surfactant. Traditionally, some authors use this equation in the form proposed by Levich [35]. However, it works only in the case of a nondeformable free surface. For a deformable surface, the equation modified by Wong et al. [36,37] needs to be used.

Let us describe the free-liquid interface as $z = h(x, y, t)$. Then, in the general case, it is necessary to distinguish between the liquid velocity $\mathbf{v}$ and the velocity along the surface, $\mathbf{v}_s = \mathbf{v} - (\mathbf{v} \cdot \mathbf{n})\mathbf{n}$. Here, $\mathbf{n} = (-\partial_x h, -\partial_y h, 1)(1 + (\partial_x h)^2 + (\partial_y h)^2)^{-1/2}$ is the unity vector

normal to the interface. Then, the evolution equation for the surfactant concentration $\Gamma$ can be written as

$$\partial_t \Gamma - \partial_h(\mathbf{e}_z \cdot \nabla_s)\Gamma + \nabla_s \cdot (\mathbf{v}_s \Gamma) + (\nabla_s \cdot \mathbf{n})(\mathbf{v} \cdot \mathbf{n})\Gamma = D_0 \nabla_s^2 \Gamma, \tag{1}$$

where $\nabla_s = \nabla - \mathbf{n}(\mathbf{n} \cdot \nabla)$ denotes the operator of the gradient along the interface, $\nabla = (\partial_x, \partial_y, \partial_z)$, $\mathbf{e}_z = (0, 0, 1)$. The first two terms of Equation (1) describe the temporal change of $\Gamma$ *along the normal* to the moving interface. The last term in the left-hand side describes the change in the surfactant concentration due to the temporal change in the surface area [38].

### 2.2. The Case of Perturbations with $k \sim O(Bi^{1/4})$

It turns out that in the case $Bi \ll 1$, there are two different characteristic spatial scales of disturbances, $k = O(Bi^{1/4})$ and $k = O(Bi^{1/2})$. The former scaling has been discovered in the pioneering work of Pearson [3] for a nondeformable surface. It was shown that in the case $Bi = O(\epsilon^4)$, $\epsilon \ll 1$, the minimum of the neutral curve is shifted to the longwave region, $k = O(\epsilon)$. We rescale the wavenumber as $k = \epsilon K$. The analysis of the Marangoni convection with the insoluble surfactant adsorbed at the deformable free surface has been conducted in [39]. That analysis based on the longwave expansions shows the existence of two modes of instability—monotonic and oscillatory. At the leading order, the critical value of the monotonic Marangoni number is

$$M_{m0} = \frac{12G(4L + N)}{L(G + 72)}. \tag{2}$$

The monotonic mode exists if

$$N < N_* = \frac{4(216 + G(G + 27))L^2}{L(864 + G(G + 36)) - G(G + 72)}.$$

The second-order correction has the form $M_{m2} = c_1 K^2 + c_2 K^{-2}$, where $c_1$ and $c_2$ are constants. For $N > N_*$, the instability becomes oscillatory with the critical Marangoni number

$$M_{osc} = \frac{1}{2(G + 27 + 48L)}\left(G^2 + 3G(41 + 32L + 5N) + 72[5N + 3 + 2L(L + N + 2)] - \sqrt{D}\right), \tag{3}$$

where

$$D = -144\{4(216 + G(G + 27))L^2 - G(G + 72)N + (864 + G(G + 36))LN\} + \\ + \{G^2 - 9G(N - 3) + 72(N + 3 + 2L(L + N))\}^2 \quad .$$

The neutral oscillations have the frequency $\omega = K^2 \Omega_0$ with

$$\Omega_0^2 = \frac{1}{288}\{-G^2 + 9G(N - 3) - 72(3 + N + 2L(L + N)) + \sqrt{D}\}.$$

The deformability of the free surface is restricted by the conservation law of the volume that gives us the equation for the local layer thickness $H$. Two others relate to the conservation of energy described by the equation for the bottom temperature $F$ and to the conservation of the mass of the surfactant described by the equation for the concentration of surfactant $\Gamma$. If all the parameters $\Sigma$, $M$, and $G$ are of order of unity, at the leading order, one obtains the following system:

$$\partial_\tau H = \nabla \cdot \left(\frac{G}{3}H^3 \nabla H + \frac{M}{2}H^2 \nabla \theta + \frac{N}{2}H^2 \nabla \Gamma\right) \equiv \nabla \cdot \tilde{\mathbf{Q}}_1, \tag{4}$$

$$H\partial_\tau F = \nabla \cdot \left(\frac{GH^4}{8}\nabla H + \frac{MH^3}{6}\nabla \theta + \frac{NH^3}{6}\nabla \Gamma + H\nabla F\right) + \tilde{\mathbf{Q}}_1 \cdot \nabla \theta - \frac{1}{2}(\nabla H)^2, \tag{5}$$

$$\partial_\tau \Gamma = \nabla \cdot \left[L\nabla \Gamma + \Gamma H\left(\frac{G}{2}H\nabla H + M\nabla \theta + N\nabla \Gamma\right)\right] \equiv \nabla \cdot \tilde{\mathbf{Q}}_2. \tag{6}$$

Here, $\theta = F - H$ is a temperature perturbation on the free surface, $\nabla = (\partial_X, \partial_Y)$. Note that parameters $\Sigma$ and $Bi$ do not appear in the leading-order equations. The obtained longwave system of equations is ill-posed [40], because the stabilizing action of the surface tension is not taken into account. To regularize the obtained problem, it is necessary to assume that the surface tension is strong: $\Sigma = \epsilon^{-2}S$, $S = O(1)$. The strong surface tension suppresses shortwave disturbances, and the assumption of longwave instability is self-consistent.

The case of the finite value of the Biot number was considered in the problem on evaporation of the liquid layer with an insoluble surfactant [37], as well as in the problem of the formation of Faraday waves at the surfactant-covered free surface of a vertically vibrated liquid layer [41].

The influence of surfactants on convective instabilities in two-layer systems was discussed in [42–44].

### 2.3. The Case of Perturbations with $k \sim O(Bi^{1/2})$

Another distinguished limit of the problem was discovered by Shklyaev et al. [13], where the authors showed that for the deformable interface, the disturbances in the interval $k \sim Bi^{1/2}$ without surfactant had two modes of instability, monotonic and oscillatory, that can be generated. This interval of wavenumbers determines the appropriate scaling for the development of possible nonlinear longwave structures in the liquid. One can rescale the spatial coordinates as

$$X = \epsilon x, \qquad Y = \epsilon y, \quad 0 < \epsilon \ll 1.$$

The characteristic time is proportional to $k^2$ and one can rescale the temporal coordinate as

$$\tau = \epsilon^2 t.$$

In the presence of an insoluble surfactant, in the framework of that scaling, we derived the following system of the longwave amplitude equations for the local thickness $H(X, Y, \tau)$, perturbations of temperature $F(X, Y, \tau)$, and surfactant concentration $\Gamma(X, Y, \tau)$, see [45]:

$$\partial_\tau H = \nabla \cdot \left( \tfrac{H^3}{3} \nabla R + \tfrac{MH^2}{2} \nabla \theta + \tfrac{NH^2}{2} \nabla \Gamma \right) \equiv \nabla \cdot \mathbf{Q_1}, \tag{7}$$

$$H \partial_\tau F = \nabla \cdot \left( \tfrac{H^4}{8} \nabla R + \tfrac{MH^3}{6} \nabla \theta + \tfrac{NH^3}{6} \nabla \Gamma + H \nabla F \right) + \mathbf{Q_1} \cdot \nabla \theta - \tfrac{1}{2} (\nabla H)^2 - \beta \theta, \tag{8}$$

$$\partial_\tau \Gamma = \nabla \cdot \left[ \Gamma H \left( \tfrac{H}{2} \nabla R + M \nabla \theta + N \nabla \Gamma \right) + L \nabla \Gamma \right] \equiv \nabla \cdot \mathbf{Q_2}. \tag{9}$$

Here, $R = GH - S\nabla^2 H$, $\nabla = (\partial_X, \partial_Y)$. The Biot number is $Bi = \epsilon^2 \beta$.

The system (7)–(9) describes the nonlinear dynamics of longwave perturbations and includes the following effects: Equation (7) includes the evolution of the surface deformation due to the bulk flow with the flow rate $\mathbf{Q_1}$ generated by hydrostatic and Laplace pressures and both thermocapillary and solutocapillary effects. Equation (8) describes the advective heat transfer by the flow and heat conductivity in the longitudinal direction, and the last two terms of the equation describe the heat loss from the deformable interface. Finally, Equation (9) presents the evolution of the surfactant concentration due to the diffusion and surface flow caused by the gravity and surface tension, as well as by thermo- and solutocapillarity.

The linear analysis of (7)–(9) was performed in [45] and gave two instability modes: the monotonic one and oscillatory one. The wavelength of disturbances is rescaled as $K = \epsilon^{-1}k$. The neutral stability curve is described for the monotonic mode as

$$M_m(K) = \frac{(48 + 12N/L)(G + K^2 S)(\beta + K^2)}{K^2(72 + G + K^2 S)}. \tag{10}$$

Formula (2) is recovered in the limit $K^2 \to \infty$, $K^2 S \to 0$. The minimum of the marginal stability curve is reached at the critical wavenumber:

$$(K_c)^2 = \frac{GS\beta + \sqrt{72GS\beta(72 + G - S\beta)}}{S(72 - S\beta)}. \tag{11}$$

Note that without the loss of generality, one can choose $S = 1$, which corresponds to the definition $\epsilon = \Sigma^{-1/2}$.

In both limits, we arrive at a conclusion that in the considered problems, despite the appearance of a new oscillatory in the mode, increasing the concentration of the insoluble surfactant suppresses the instabilities creating the convection.

## 3. Pattern Selection in the Longwave Marangoni Convection

### 3.1. Square Lattice

We consider the nonlinear dynamics of small spatially periodic perturbations in the neighborhood of the threshold of the monotonic mode, $M = M_m + \delta^2 M_2$ (here, $\delta$ is a small parameter of supercriticality), corresponding to a square lattice in the Fourier space. At the present stage, only disturbances with the critical wavevector $K_c$ determined by (11) are taken into account. The plots showing basic wavevectors for all considered lattices are shown in Appendix A.

The problem is described by the set of Equations (7)–(9) for variables $H$, $F$, and $\Gamma$, which are expanded in powers of parameter $\delta$:

$$\begin{aligned}
H &= 1 + \delta h_1 + \delta^2 h_2 + \dots, \\
F &= 1 + \delta f_1 + \delta^2 f_2 + \dots, \\
\Gamma &= 1 + \delta \gamma_1 + \delta^2 \gamma_2 + \dots.
\end{aligned} \tag{12}$$

Substituting these expansions into (7)–(9), we obtain at the leading order the linear equations for $h_1$, $f_1$, and $\gamma_1$. We can write their solution in the form

$$h_1 = A_1(\tau_2)e^{iK_c X} + A_2(\tau_2)e^{iK_c Y} + c.c., \quad f_1 = \alpha_1 h_1, \quad \gamma_1 = \alpha_2 h_1, \tag{13}$$

where $\alpha_1 = -\frac{(GK^2 + K^4 S - 72\beta)}{72(K^2 + \beta)}$ and $\alpha_2 = \frac{G + K^2 S}{6L}$.

Continuing the operations for higher powers of $\delta$ and applying the solvability conditions at the third order in $\delta$, we obtain the set of Landau equations that describe the evolution of complex amplitudes $A_1$ and $A_2$:

$$\frac{\partial A_1}{\partial \tau_2} = \kappa_0 A_1 + \kappa_1 |A_1|^2 A_1 + \kappa_2 |A_2|^2 A_1, \tag{14}$$

$$\frac{\partial A_2}{\partial \tau_2} = \kappa_0 A_2 + \kappa_1 |A_2|^2 A_2 + \kappa_2 |A_2|^2 A_2. \tag{15}$$

Here,

$$\kappa_0 = \frac{K^4 L^2 (72 + G + K^2 S)^2 M_2}{12(K^2 D_1 - \beta D_2)} \tag{16}$$

where

$$\begin{aligned}
D_1 = {}&-N(G + K^2 S)(72 + G + K^2 S) + LN(864 + 36G + G^2 + 2(18 + G)K^2 S + K^4 S^2) + \\
&4L^2(216 + 27G + G^2 + (27 + 2G)K^2 S + K^4 S^2), \\
D_2 = {}&36LN(-24 + G + K^2 S) + N(G + K^2 S)(72 + G + K^2 S) + 36L^2(-24 + 5G + 5K^2 S).
\end{aligned}$$

Other coefficients are cumbersome to be given here (see Supplemental Material to [45]).

We have two kinds of steady-state solutions: one of them describes rolls (if one of $A_1$ or $A_2$ is zero) and another one describes squares (both $A_1$ and $A_2$ are nonzero). Here, the pattern selection is determined by signs of $\kappa_1$, $\kappa_1 - \kappa_2$, and $\kappa_1 + \kappa_2$. The rolls and squares

can be stable if $\kappa_1 < 0$ and $\kappa_1 + \kappa_2 < 0$ (supercritical bifurcation). The rolls are selected if $\kappa_2 < \kappa_1 < 0$; if $\kappa_1 < \kappa_2$, $\kappa_1 + \kappa_2 < 0$, the squares are stable, see Figure 2.

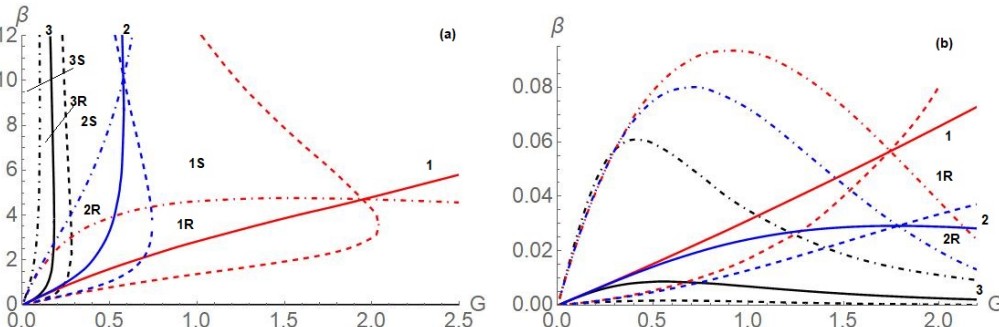

**Figure 2.** Pattern selection domains for the square lattice. The solid lines are for $\kappa_1 = 0$, the dashed lines are for $\kappa_1 + \kappa_2 = 0$, and dot-dashed lines are for $\kappa_2 - \kappa_1 = 0$. Here, the red-colored set (# 1 ) is for $N = 0$, the blue-colored set (# 2) is for $N = 10^{-6}$, and the black set (#3) is for $N = 10^{-5}$. The domains are marked: "R" for rolls, "S" for squares. Panel (**b**) is detailed panel (**a**) for small $\beta$.

Note that the problem of selection between the roll and square patterns is mathematically similar to the classical problem of species competition [46]. The selection of rolls corresponds to the extinction of all species except one, while the selection of squares corresponds to the symbiosis.

The pattern selection domains are presented in Figure 2 for three different cases. The red color is for the case without surfactant, the blue color relates to $N = 10^{-6}$, and the black color is for $N = 10^{-5}$. It is seen that even at a low concentration of surfactant, the domains shrink. Figure 2b shows domains at small values of $\beta$.

It was found that the rolls are the most typical of the patterns. Note that this kind of structures is the only possible one in a rectangular container with one size much longer than another one or in a narrow annular container of a large radius.

### 3.2. Hexagonal Lattice

The most typical kind of non-equilibrium patterns near the threshold of stationary is the hexagonal pattern. Hexagons are formed by three sets of rolls, with wavenumbers satisfying the resonant condition $\mathbf{k}_1 + \mathbf{k}_2 + \mathbf{k}_3 = 0$, $|k_1| = |k_2| = |k_3|$. Well-known examples are irregular hexagons appearing in the heated cooking oil and giraffe's coat markings (see [14]); also, hexagonal patterns were observed in front solidification [47], in Faraday crispation [48], in a liquid-crystal-valve device [49], in ferrofluids [50], etc.

Here, we find how the surfactant concentration affects the type of hexagons in the longwave Marangoni convection. As in the previous section, we take the variable fields $H(X, Y, \tau)$, $F(X, Y, \tau)$, and $\Gamma(X, Y, \tau)$ near the threshold point $M_m(K_c)$ (see (10) and (11)). The Marangoni number is presented as $M = M_m + \delta M_1 + \delta^2 M_2 + \ldots$. We introduce the different time scales in the neighborhood of the threshold, i.e., $\tau_0 = \tau$, $\tau_1 = \delta\tau$, $\tau_2 = \delta^2\tau_2$, etc. Then, at the leading order of the expansions, at the hexagonal lattice in the Fourier space, we have the solution

$$h_1 = \sum_{j=1}^{3} A_j(\tau_1, \tau_2)e^{iK_c\mathbf{n_j}\cdot\mathbf{X}} + c.c, \quad f_1 = \alpha_1 h_1, \quad \gamma_1 = \alpha_2 h_1. \tag{17}$$

Expression (17) corresponds to six peaks in the Fourier space around the points $\pm K_c\mathbf{n_j}$ ($j = 1, 2, 3$), $\mathbf{n}_1 = (1, 0)$, $\mathbf{n}_2 = (-1/2, \sqrt{3}/2)$, $\mathbf{n}_3 = (-1/2, -\sqrt{3}/2)$. The vectors $\mathbf{n}_j$ are unit vectors parallel to pattern wavevectors. We also define mutually orthogonal to $\mathbf{n}_j$ unit vectors $\mathbf{t_j}$, namely $\mathbf{t}_1 = (0, 1)$, $\mathbf{t}_2 = (-\sqrt{3}/2, -1/2)$, $\mathbf{t}_3 = (\sqrt{3}/2, -1/2)$. Here, $\mathbf{X} = (X, Y)$.

At the second order of the expansion, the solvability condition for $h_1$, $f_1$, and $\gamma_1$ gives

$$\partial_{\tau_1} A_l = \kappa_0 A_l + s_1 A_m^* A_n^*, \quad \{l, m, n\} = \{1, 2, 3\}, \{2, 3, 1\}, \{3, 1, 2\}. \tag{18}$$

If $s_1 = O(1)$, then Equation (18) describes a two-sided bifurcation and unbounded growth of the selected type of hexagons, *up-hexagons* or *down-hexagons*, depending on the sign $s_1$. However, there exists the line in the space of parameters, where $s_1 = 0$; hence, the term $s_1 A_m^* A_n^*$ vanishes. The line $s_1(G, \beta) = 0$ is shown in Figure 3. The solid line corresponds to the case without the surfactant, and the dashed line is plotted in the case with added surfactant on the surface ($N = 10^{-6}$ and $L = 0.003$). In all the calculations, here we choose $S = 1$, which corresponds to the definition $\epsilon \equiv \Sigma^{-1/2}$.

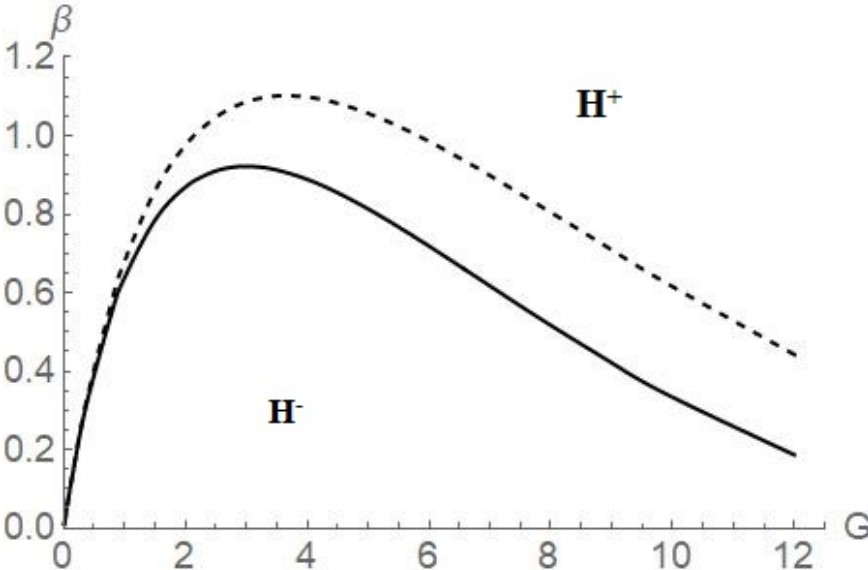

**Figure 3.** Pattern selection for regular hexagons ($q_0 = 0$). Solid line: $s_1(G, \beta) = 0$ in the case without surfactant ($N = 0$), the dashed line: in the case with surfactant ($N = 10^{-6}$, $L = 0.003$). Up-hexagons ($H^+$) are selected in the region above the line $s_1 = 0$, and down-hexagons ($H^-$) are selected below that line.

In the region where $s_1 = O(\delta)$, we can denote $s_1 = \delta \bar{s}_1$ and relegate this term to the third order of expansion. In the vicinity of the line $s_1 = 0$, we can set $\partial_{\tau_1} = M_1 = 0$. Then, we obtain equations:

$$\partial_{\tau_2} A_l = \kappa_0 A_l + \kappa_1 |A_l|^2 A_l + \kappa_2 (|A_m|^2 + |A_n|^2) A_l + \bar{s}_1 A_m^* A_n^*,$$
$$\{l, m, n\} = \{1, 2, 3\}, \{2, 3, 1\}, \{3, 1, 2\}, \tag{19}$$

where $\kappa_1$ is the same as in Equations (14) and (15), and $\kappa_2$ characterizes the interaction of rolls turned to the angle $2\pi/3$ with each other. The growth of amplitudes is saturated if $\kappa_1 < 0$ and $\kappa_1 + 2\kappa_2 < 0$.

### 3.3. Rhombic Lattice

The patterns of a rhombic lattice are less widespread than rolls/squares or hexagons. However, they are known in different physical processes, such as [51–54]. These patterns are the result of the interaction of two external perturbations, oriented by angle $\theta$ one to another. In other words, we have an external one in rolls with respect to disturbances such that their wavevectors form angle $\theta$ with the basic wavevector of the roll. Thus, the solutions at the leading order of the variables can belong to a rhombic lattice in the Fourier space:

$$h_1 = A_1(\tau_2)e^{iK_cX} + A_2(\tau_2)e^{i(K_cX\cos\theta + K_cY\sin\theta)} + c.c., \quad f_1 = \alpha_1 h_1, \quad \gamma_1 = \alpha_2 h_1. \tag{20}$$

The angle $\theta$ is an additional parameter of the problem. Without a loss of generality, we take for consideration $\theta < \pi/2$, but $\theta$ does not equal $\pi/3$. The rhombic lattice at $\theta = \pi/2$ becomes a square one. Note that patterns with wavevectors on the rhombic lattice in the Fourier space look like *rectangles* in the real space.

Making the same procedures as in the case of the square lattice, we obtain for the complex amplitudes $A_1$ and $A_2$ the system of equations similar to (14) and (15):

$$\frac{\partial A_1}{\partial \tau_2} = \kappa_0 A_1 + \kappa_1 |A_1|^2 A_1 + \tilde{\kappa}_2(\theta)|A_2|^2 A_1, \tag{21}$$

$$\frac{\partial A_2}{\partial \tau_2} = \kappa_0 A_2 + \kappa_1 |A_2|^2 A_2 + \tilde{\kappa}_2(\theta)|A_2|^2 A_2. \tag{22}$$

Here, $\kappa_2 = \tilde{\kappa}_2(\pi/2)$. The growth of amplitudes on the rhombic lattice is saturated if $\kappa_1 < 0$ and $\kappa_1 + \tilde{\kappa}_2 < 0$.

Figure 4 presents the pattern selection on the rhombic lattice for three values of angle $\theta$. Figure 4a shows the case $\theta = 0.45\pi$. Here, the angle is closer to $\pi/2$ and the map is similar to the patterns on the square lattice. In Figure 4b is the case $\theta = 0.4\pi$, and in Figure 4c, $\theta = 0.25\pi$.

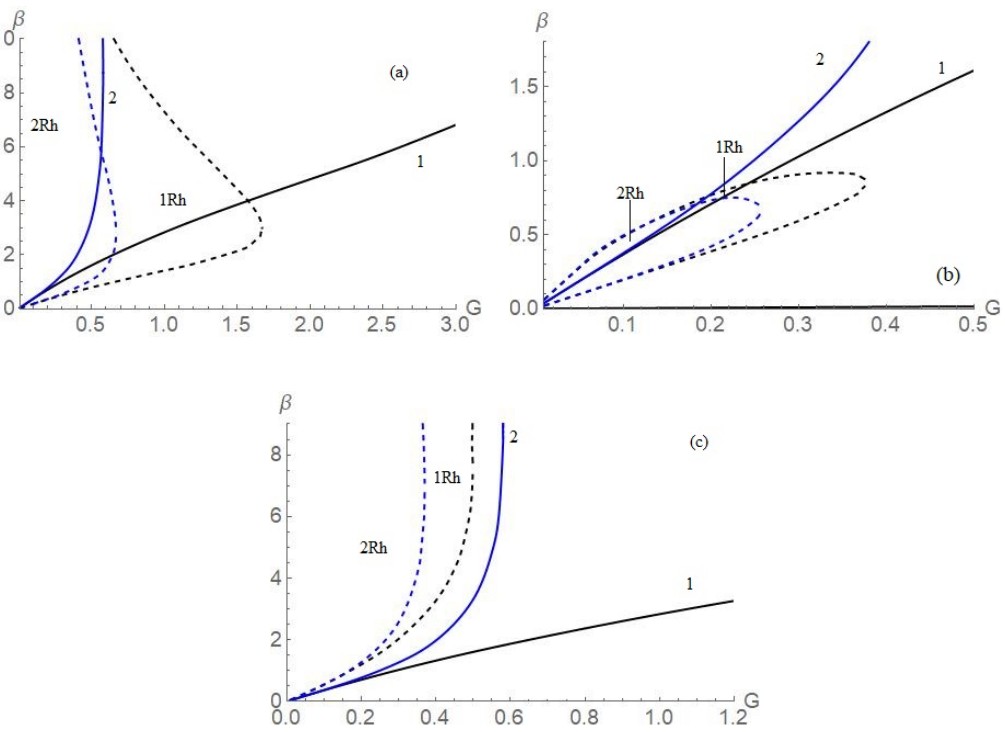

**Figure 4.** Pattern selection domains for the rhombic lattice. The solid lines are for $\kappa_1 = 0$, the dashed lines are for $\kappa_1 + \tilde{\kappa}_2(\theta) = 0$. Here, the black color set (#1 ) is for $N = 0$, the blue-colored set (#2 ) is for $N = 10^{-6}$. The stability domains are marked: "1Rh" is for $N = 0$, "2Rh" for $N = 10^{-6}$. Panel (**a**): $\theta = 0.45\pi$, panel (**b**): $\theta = 0.4\pi$, and panel (**c**): $\theta = 0.25\pi$.

Each panel includes two cases of different $N$; the black color lines are for $N = 0$ and the blue ones are for $N = 10^{-6}$. It is shown that surfactant diminishes the regions of stable rhombi. The calculations show that the most dangerous perturbations are directed by $\theta = \pi/2$.

## 4. Modulational Instability of Stationary Rolls—The Case of 1D Disturbances

The stability analysis carried out in the previous section is incomplete because it does not include the stability of periodic patterns with respect to disturbances with wavenumbers different from $K_c$. The most important class of disturbances are those with wavenumbers close to $K_c$. Below, we present the analysis of the instability of roll patterns with respect to one-dimensional disturbances near the instability threshold ($M = M_0 + \delta^2 M_2$, $M_0 = M_m(K_c)$) carried out in [31]. In that case, the motionless state is unstable with respect to disturbances with the wavenumbers in the interval $O(\delta)$ around $K_c$. Using the Newell–

Whitehead–Segel approach, two spatial scales are introduced: $X$ corresponding to the main wavelength of the pattern, and $X_1 = \delta X$ corresponding to the spatial modulation of the pattern. The time variable is rescaled as $\tau_2 = \delta^2 \tau$. Thus, the solution at the leading order can be written as

$$h_1 = A(X_1, \tau_2)e^{iK_cX_0} + A(X_1, \tau_2)^* e^{-iK_cX_0}, f_1 = \alpha_1 h_1, \gamma_1 = \alpha_2 h_1$$

(coefficients $\alpha_1$ and $\alpha_2$ are the same as in the previous section). At the next order, the solution can be written as

$$
\begin{aligned}
h_2 &= \bar{h}(X_1, \tau_2) + B_1(X_1, \tau_2)e^{2iK_cX_0} + B_1(X_1, \tau_2)^* e^{-2iK_cX_0}, \\
f_2 &= \bar{f}(X_1, \tau_2) + B_2(X_1, \tau_2)e^{2iK_cX_0} + B_2(X_1, \tau_2)^* e^{-2iK_cX_0}, \\
\gamma_2 &= \bar{\gamma}(X_1, \tau_2) + B_3(X_1, \tau_2)e^{2iK_cX_0} + B_3(X_1, \tau_2)^* e^{-2iK_cX_0}.
\end{aligned}
\tag{23}
$$

The heat-transfer equation at the second order gives $\bar{f} - \bar{h} = -\frac{K_c^2}{\beta}|A|^2$. The calculation is performed taking into account that convection does not change the total volume of the liquid and the total amount of the surfactant one has $< \bar{h} >= 0, < \bar{\gamma} >= 0, (< \cdots >$ is averaging over $X_1$).

Finally, one obtains the following set of equations for the amplitude of the rolls $A$, for the large-scale disturbances of the surface deformation $h_0$, and for surfactant concentration $\gamma_0$:

$$\partial_{\tau_2} A = \kappa_0 A + \kappa_1 |A|^2 A + \mu_1 \partial_{X_1 X_1} A + \mu_2 \bar{h} A + \mu_3 \bar{\gamma} A, \tag{24}$$

$$\partial_{\tau_2} \bar{h} = a_1 \partial_{X_1 X_1} |A|^2 + a_2 \partial_{X_1 X_1} \bar{h} + a_3 \partial_{X_1 X_1} \bar{\gamma}, \tag{25}$$

$$\partial_{\tau_2} \bar{\gamma} = b_1 \partial_{X_1 X_1} |A|^2 + b_2 \partial_{X_1 X_1} \bar{h} + b_3 \partial_{X_1 X_1} \bar{\gamma}. \tag{26}$$

The nature of the terms and the expressions for the coefficients are described in [31]. The coefficients $\mu_i (i = 1, 2, 3)$ are cumbersome and not presented here. Coefficients $a_i$ and $b_i$ are described in Appendix B.

Stationary solution of Equations (24)–(26) without deformation and non-homogeneity of surfactant is a family of rolls $A_0 = \sqrt{\frac{q^2 \mu_1 - \kappa_0}{\kappa_1}} e^{iqX_1}$, $\bar{h} = 0$, $\bar{\gamma} = 0$ $(\kappa_1 < 0)$. Here, $q$ is the deviation of the wavenumber from $K_c$ $(q^2 < q_0^2 = \kappa_0 / \mu_1)$.

To consider the stability of the rolls, we disturb their amplitude and the phase in the form of normal modes. Obtaining the dispersion relation for the growth rate, $\lambda$, of the side-band instability, we find in the limit of longwave modulations (assuming $\lambda = \lambda_0 + \Lambda k^2 + o(k^2)$) that the growth rate of the mode related to the amplitude modulation, $\lambda_0$, is negative and the other three eigenvalues tend to zero when $k \to 0$. This equation for $\Lambda$ describes the interaction of three Goldstone modes corresponding to the definite symmetries of the problem—phase disturbances, surface deformation, and surfactant concentration disturbances.

The analysis of the dispersion relation in the case without surfactant gives a quadratic equation for eigenvalue $\Lambda_{1,2}$. Fixing parameters $G$ and $\beta$ (remember, $S = 1$), we can follow the evolution of the modulated rolls as a function of $q^2$.

Figure 5 presents the stability map of the stationary rolls in the case without surfactant. The black solid line here shows the boundary between supercritical and subcritical domains for the rolls. We analyzed the modulational instability only in the region of supercritical rolls and found three regions:

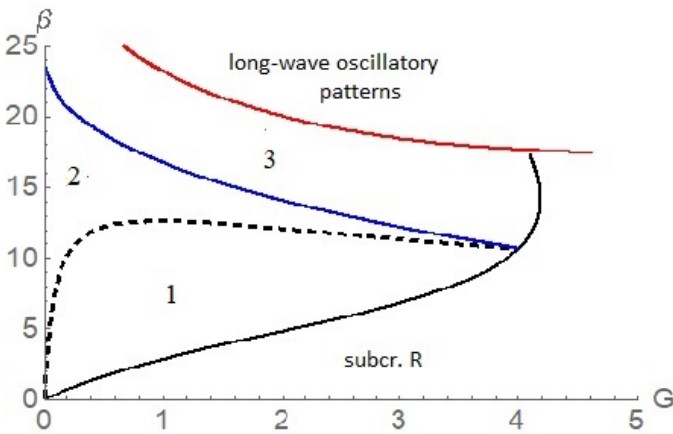

**Figure 5.** Stability map of the stationary rolls in the case without surfactant.

- Region 1. Here, all the rolls are unstable with respect to modulation.
- Region 2. Here, the rolls are stable within the interval $q^2 < q_m^2$ and monotonically unstable for $q_m^2 < q^2 < q_0^2$ (monotonic Eckhaus instability).
- Region 3. Here, the rolls are stable within interval $q^2 < q_{osc}^2$, oscillatory unstable for $q^2$ slightly above $q_{osc}^2$, and oscillatory or monotonically unstable for $q_{osc}^2 < q^2 < q_0^2$.

The boundary of the monotonic Eckhaus instability is found as

$$q_m^2 = \frac{\kappa_0}{\mu_1} \left( \frac{a_2 \kappa_1 - a_1 \mu_2}{3 a_2 \kappa_1 - a_1 \mu_2} \right). \tag{27}$$

For $q_{osc}^2$, we found

$$q_{osc}^2 = \frac{\kappa_0}{\mu_1} \left( \frac{a_2 \kappa_1 + \kappa_1 \mu_1 - a_1 \mu_2}{a_2 \kappa_1 + 3 \kappa_1 \mu_1 - a_1 \mu_2} \right). \tag{28}$$

Even a small addition of surfactant changes the stability interval. The boundary of monotonic Eckhaus instability is

$$q_m^2 = q_0^2 \frac{1 + C}{3 + C}, \tag{29}$$

where $q_0^2 = \kappa_0 / \mu_1$ is the width of the existence interval for periodic solutions and

$$C = \frac{(a_3 b_1 - a_1 b_3) \mu_2 - (a_2 b_1 - a_1 b_2) \mu_3}{(a_2 b_3 - a_3 b_2) \kappa_1}. \tag{30}$$

In the case without surface deformations and disturbances of surfactant concentration, we have $C = 0$, and we obtain the classical result $q_m^2 = (1/3) q_0^2$.

Figure 6 presents the stability maps for $N = 10^{-8}$ (**a**) and $N = 10^{-7}$ (**b**). The shrinking of the region of stable supercritical rolls means that with an increasing surfactant concentration, the instability becomes subcritical in the larger region of the parameter.

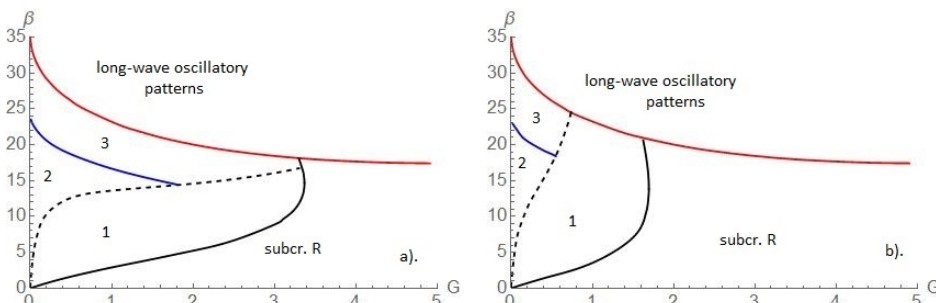

**Figure 6.** Stability maps for $N = 10^{-8}$ (**a**) and $N = 10^{-7}$ (**b**). Other parameters $L = 0.003$ and $S = 1$. Regions "1", "2", and "3" correspond to the same named in Figure 5.

## 5. Modulational Instability of Stationary Rolls—The Case of 2D Disturbances

Consider now the 2D modulation of the rolls. The roll pattern is typically selected by the analysis of competition of disturbances characterized by difference $|\mathbf{K_n} - \mathbf{K_m}| \sim O(1)$, i.e., the angle between these two wavevectors is $O(1)$. In the case of the problem with rotational symmetry, the set of wavevectors corresponding to unstable modes is a ring which contains a continuum of wavevectors with $|K^2 - 1| < \delta$, see Figure 7.

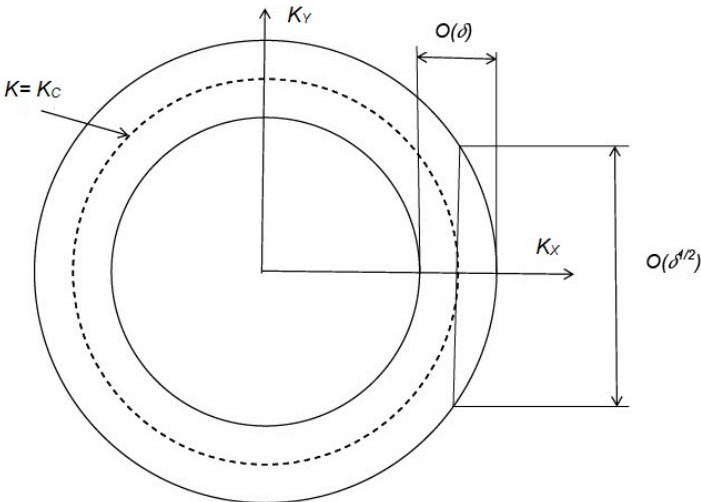

**Figure 7.** Ring of unstable wavevectors.

Let us take a roll pattern with $\mathbf{K}_0 = (K_c, 0)$ and consider the disturbances with $\mathbf{K} = \mathbf{K}_0 + \boldsymbol{\Delta}$ around that wavevector. The most unstable modes have $K = K_c$. When $Ma - Ma_c \sim O(\delta^2)$, the instability region of the motionless state is a ring $||\mathbf{K}| - K_c| = O(\delta)$. Because

$$|\mathbf{K}| = |\mathbf{K} + \Delta| = \sqrt{(K_c + \Delta_x)^2 + \Delta_y^2} = K_c + \Delta_x + \frac{\Delta_y^2}{2K_c} + \dots,$$

the instability of the motionless state takes place with respect to the disturbances with the wavevectors in the region

$$|\Delta_x| = O(\delta), \qquad |\Delta_y| = O(\delta^{1/2}). \tag{31}$$

In addition, the coupling between the longwave roll modulation, the modulations of the surface distortion $h$, and the distribution of the surfactant concentration $\gamma$ has some specific features in the region of $|K_y| = O(\delta)$. Therefore, one has to consider two regions of transversal modulation characterized by rescaled variables $Y_{1/2} = \delta^{1/2}Y$ and $Y_1 = \delta Y$.

*5.1. Transversal Modulation of Rolls, $Y_1 = \delta Y$*

Introduce the following rescaling of spatial and time coordinates

$$X_0 = X, \quad X_1 = \delta X, \quad Y_1 = \delta Y, \quad \tau_2 = \delta^2 \tau. \tag{32}$$

As in the previous section, applying the approach of Newell–Whitehead–Segel, one obtains the following set of evolution equations for amplitude $A$, for the surface deformation $h_0$, and for the disturbance of surfactant concentration $\gamma_0$:

$$\partial_{\tau_2} A = \kappa_0 A + \kappa_1 |A|^2 A + \mu_1 \partial_{X_1 X_1} A + \mu_2 h_0 A + \mu_3 \gamma_0 A, \tag{33}$$

$$\partial_{\tau_2} h_0 = (a_1 \partial_{X_1 X_1} + \tilde{a}_1 \partial_{Y_1 Y_1})|A|^2 + a_2 (\partial_{X_1 X_1} + \partial_{Y_1 Y_1}) h_0 + a_3 (\partial_{X_1 X_1} + \partial_{Y1Y1}) \gamma_0, \tag{34}$$

$$\partial_{\tau_2} \gamma_0 = (b_1 \partial_{X_1 X_1} + \tilde{b}_1 \partial_{Y_1 Y_1})|A|^2 + b_2 (\partial_{X_1 X_1} + \partial_{Y1Y1}) h_0 + b_3 (\partial_{X_1 X_1} + \partial_{Y_1 Y_1}) \gamma_0. \tag{35}$$

Coefficients $\kappa_1$, $\mu_1$, $\mu_2$, and $\mu_3$ are cumbersome and not given here; others are presented in Appendix A.

In the framework of the linear stability theory, the perturbations of the roll amplitude and its phase, surface deviation, and surfactant of concentration are taken proportional to $e^{\lambda \tau_2 + i K_X X_1 + i K_Y Y_1}$. Consider a longwave modulation of the rolls and

$$|\mathbf{k}|^2 = k^2 = K_X^2 + K_Y^2 = K_X^2 (1 + \chi^2) \ll 1.$$

Here, $\chi = K_Y / K_X$. When $|K_Y| = 0$, we have the case described in the previous section. The boundary of the monotonic Eckhaus instability is described by (29). In the case of transversal modulation, $|K_Y| \neq 0$, the parameter $C$ in (29) is replaced by $\tilde{C} = \frac{C + \tilde{C}\chi^2}{1 + \chi^2}$. Here, $\tilde{C}$ is obtained from $C$ by the replacement $a_1 \to \tilde{a}_1$ and $b_1 \to \tilde{b}_1$. Because for any $\chi^2$, $\tilde{C}$ is between $C$ and $\tilde{C}$, it is sufficient to compare the criterion with the new one,

$$q^2 < \tilde{q}_m^2 = q_0^2 \frac{1 + \tilde{C}}{3 + \tilde{C}}. \tag{36}$$

The rolls are stable with respect to disturbances with small $K_X^2$, $K_Y^2$, if $q^2 < \min(q_0^2, \tilde{q}_m^2)$. The stability maps have additional parameter, $\chi$. We start plot for case without the surfactant, see Figure 8. When $\chi$ grows, the stability of stationary rolls increase, i.e., the longitudinal disturbances are more dangerous than disturbances with inclined wavevector. At $\chi^2 > 5.2$, the region "1" disappears.

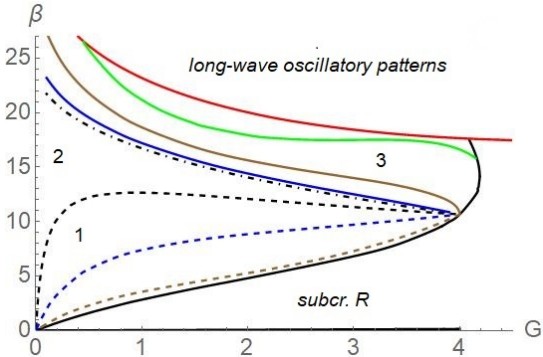

**Figure 8.** Stability map of the stationary rolls in the case without surfactant. The solid black line is the boundary line between supercritical and subcritical rolls ("subcr. R"). Region "1" (below the dashed line) is the region of unstable rolls; in region "2", rolls are stable within $q^2 < q_m^2$ and monotonically unstable for $q_m^2 < q^2 < q_0^2$; region "3" (above dot-dashed line) is the region of stable rolls for $q^2 < q_{osc}^2$, oscillatory unstable for $q^2$ slightly above $q_{osc}^2$, and oscillatory or monotonically unstable for any $q_{osc}^2 < q^2 < q_0^2$. Dashed and solid lines: blue color for $\chi^2 = 2$, brown color for $\chi^2 = 5$, green color for $\chi^2 = 10$.

Figure 9 presents the stability maps for two different values of $N$: panel (a) $N = 10^{-8}$, panel (b) $N = 10^{-7}$ for different $\chi$.

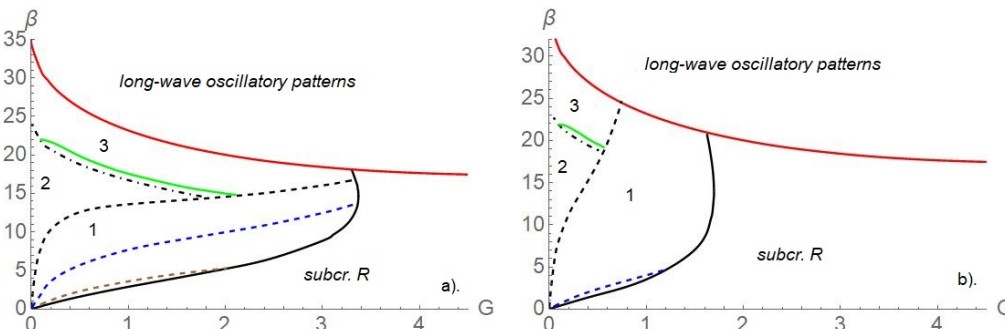

**Figure 9.** Stability maps for $N = 10^{-8}$ (**a**) and $N = 10^{-7}$ (**b**). Other parameters are $L = 0.003$ and $S = 1$. Regions "1", "2", and "3" are the same as Figure 3. Dashed and solid lines: blue color for $\chi^2 = 2$, brown color for $\chi^2 = 5$, green color for $\chi^2 = 10$.

*5.2. Transversal Modulation of Rolls, $Y_{1/2} = \delta^{1/2} Y$*

Here, we rescale coordinates as

$$X_0 = X, \quad X_1 = \delta X, \quad Y_{1/2} = \delta^{1/2} Y, \tau_2 = \delta^2 \tau. \tag{37}$$

Now, Equations (34) and (35) in the third order become

$$\partial^2_{Y_{1/2}} (\tilde{a}_1 |A|^2 + a_2 h_0 + a_3 \gamma_0), \tag{38}$$

$$\partial^2_{Y_{1/2}} (\tilde{b}_1 |A|^2 + b_2 h_0 + b_3 \gamma_0). \tag{39}$$

Thus, the system of algebraic equations for surface distortion $h_0$ and for distribution of the surfactant concentration $\gamma_0$ can be written as

$$a_2 h_0 + a_3 \gamma_0 = -\tilde{a}_1 (|A|^2 - < |A|^2 >), \tag{40}$$

$$b_2 h_0 + b_3 \gamma_0 = -\tilde{b}_1 (|A|^2 - < |A|^2 >), \tag{41}$$

where $< \cdots >$ denotes averaging over both $X_1$ and $Y_{1/2}$. Solving this system, we find the solutions for surface distortion $h_0$ and distribution of the surfactant concentration $\gamma_0$ can be written as

$$h_0 = c_h (|A|^2 - < |A|^2 >), \qquad \gamma_0 = c_\gamma (|A|^2 - < |A|^2 >). \tag{42}$$

Here,

$$c_h = \frac{a_3 \tilde{b}_1 - \tilde{a}_1 b_3}{a_2 b_3 - b_2 a_3}, \quad c_\gamma = \frac{\tilde{a}_1 b_2 - a_2 \tilde{b}_1}{a_2 b_3 - b_2 a_3}.$$

In this case, we can rewrite the amplitude equation as the following modified Newell–Whitehead–Segel equation:

$$\partial_{\tau_2} A = \kappa_0 A + \kappa_1 |A|^2 A + \tilde{\kappa}_1 (|A|^2 - < |A|^2 >)A + \mu_1 \left( \frac{\partial}{\partial X_1} - \frac{i \partial^2}{2 K_c \partial Y_{1/2}^2} \right)^2 A. \tag{43}$$

Here, $\tilde{\kappa}_1 = \mu_2 c_h + \mu_3 c_\gamma$. Linearizing Equation (43) and substituting the disturbances of the rolls amplitude and its phase in the form $\sim \exp(\lambda \tau_2 + i(K_X X_1 + K_Y Y_{1/2}))$, we obtain quadratic equation for the growth rate $\lambda$.

In the case when the perturbations vary only in $Y-$direction (parallel to the rolls), $K_X = 0$ the dispersion relation gives the growth rate for the phase perturbation

$$\lambda = -K_Y^2 \frac{q\mu_1}{K_c} - K_Y^4 \frac{\mu_1}{4K_c^2}. \tag{44}$$

Here, when $q < 0$, we have the zigzag instability for small $|K_Y|$ ($K_Y^2 < -4K_c q$). The zigzag instability caused by the phase disturbances is not influenced by $\tilde{\kappa}_1$. The origin of the zigzag instability is fully "geometrical" for $K < K_c$, there is a resonance of neutral disturbances with wavevectors $(\pm K, K_Y)$ with zero linear growth rate, $K^2 + K_Y^2 = K_c^2$:

$$(-K, K_Y) + (2K, 0) = (K, K_Y).$$

## 6. Distortions of Hexagons

Let us discuss now the longwave distortion of hexagons:

$$h_1 = \sum_{j=1}^{3} A_j(\mathbf{X}_1, \tau_1, \tau_2) e^{iK_c \mathbf{n_j} \cdot \mathbf{X}_0} + c.c, \quad f_1 = \alpha_1 h_1, \quad \gamma_1 = \alpha_2 h_1, \tag{45}$$

where $\mathbf{X}_0 = \mathbf{X}$, $\mathbf{X}_1 = \delta\mathbf{X}$. Distorted non-equilateral hexagons have been observed in chemical systems [55] and considered in theoretical models [56,57].

The general form of the amplitude equations for distorted patterns described by amplitudes $A_l$ ($l = 1, 2, 3$) is as follows, [58,59]:

$$\partial_{\tau_2} A_l = \kappa_0 A_l + \kappa_1 |A_l|^2 A_l + \kappa_2 (|A_m|^2 + |A_n|^2) A_l + \mu_1 (\mathbf{n_l} \cdot \nabla)^2 A_l + \mu_2 \bar{h} A_l +$$
$$+ \mu_3 \bar{\gamma} A_l + \bar{s}_1 A_m^* A_n^* + i s_2 [A_m^* (\mathbf{t_n} \cdot \nabla) A_n^* - A_n^* (\mathbf{t_m} \cdot \nabla) A_m^*] + i s_3 [A_n^* (\mathbf{n}_m \cdot \nabla) A_m^* +$$
$$+ A_m^* (\mathbf{n}_n \cdot \nabla) A_n^*], \tag{46}$$

$$\partial_{\tau_2} \bar{h} = \sum_{j=1}^{3} [a_1 (\mathbf{n}_j \cdot \nabla)^2 + \tilde{a}_1 (\mathbf{t}_j \cdot \nabla)^2] |A_j|^2 + a_2 \nabla^2 \bar{h} + a_3 \nabla^2 \bar{\gamma}, \tag{47}$$

$$\partial_{\tau_2} \bar{\gamma} = \sum_{j=1}^{3} [b_1 (\mathbf{n}_j \cdot \nabla)^2 + \tilde{b}_1 (\mathbf{t}_j \cdot \nabla)^2] |A_j|^2 + b_2 \nabla^2 \bar{h} + b_3 \nabla^2 \bar{\gamma}. \tag{48}$$

Here, $\nabla \equiv \nabla_{\mathbf{X}_1}$. New coefficients are: $\tilde{a}_1 = \frac{1}{3}(G + K_c^2 S) - \frac{K_c^2 M_m}{2\beta}$ and $\tilde{b}_1 = \frac{1}{36L}(G + K_c^2 S)(12L - G - K_c^2 S) - \frac{K_c^2 M_m}{\beta}$. The terms with the coefficient $s_2$ describe the change in the nonlinear interaction by rotation of the wavevectors of rolls forming the hexagons, while the terms with $s_3$ correspond to that by dilatations of hexagons. The values of the coefficients $s_2$ and $s_3$ are computed directly from the original Equations (7)–(9), similarly to [60,61]. The terms $\mu_2 \bar{h} A_l$ and $\mu_3 \bar{\gamma} A_l$ describe the influence of surface deformation and surfactant concentration perturbation created by the pattern modulation on the growth rate of Marangoni instability.

## 7. Conclusions

In this paper, a review is presented of the recent works on the longwave Marangoni convection of the non-isothermal liquid layer covered by an insoluble surfactant. This analysis includes two different ranges of disturbances: one is with wavevectors $k \sim Bi^{1/4}$ and one with $k \sim Bi^{1/2}$ ($Bi$ is the Biot number). While the first interval of wavevectors is well known and widely investigated, the second one, $k \sim Bi^{1/2}$, is less known. Recent works elucidate the pattern selection at the interval of the disturbance wavenumbers: rolls/squares, hexagons, and rhombi were found. In the present work, we studied how the concentration of an insoluble surfactant, characterized by the elasticity parameter $N$, influences the roll stability, as well as that of rectangles and hexagons. The last pattern must be still investigated in more detail. The analysis of external instability is performed

at a fixed value of the critical wavevector at which the monotonic instability mode of the Marangoni convection appears.

The special part of the paper is devoted to the modulational instability, when the 1D or 2D large-scale perturbations disturb the stable roll patterns, which are formed as a result of the stationary Marangoni convection in the liquid layer with/without insoluble surfactant. The result of this modulation is the appearance of the Eckhaus instability. The analysis is performed in the neighborhood of the critical wavenumber. The difference between the 1D and 2D modulation is revealed. In both cases, the stability maps are drawn in the plane of the Galileo number, $G$, and the Biot number, $Bi$.

The existence of three regions of the supercritical rolls is found. Region "1" is where all the rolls are unstable with respect to modulation. Region "2" is where the rolls are stable within the interval $q^2 < q_m^2$ and monotonically unstable for $q_m^2 < q^2 < q_0^2$. Region "3" is where the rolls are stable within the interval $q^2 < q_{osc}^2$, oscillatory unstable for $q^2$ slightly above $q_{osc}^2$, and oscillatory or monotonically unstable for any $q_{osc}^2 < q^2 < q_0^2$. The zigzag instability boundary does not depend on the concentration of the insoluble surfactant.

The paper includes the known results published earlier in different papers, as well as new results never yet published. The authors anticipate that the discussed theory can motivate more experiments on the investigation of pattern formation in the system with surfactants. To estimate the parameters of potential experiments, we take the water layer of a thickness of 0.05 mm. As shown in [13], the longwave Marangoni convection is possible if $Bi\Sigma < 72$. Assuming the heat-transfer coefficient $q \sim 10$ (W/m$^2$K) and the thermal conductivity is 0.5 (W/mK), one obtains $Bi \sim 0.001$. The Galileo number corresponds to $G \sim 10$ and the inverse capillary number $\Sigma \sim 3 \times 10^4$.

The authors hope that this paper will be useful for experts in colloid physics and in surface active agents (surfactants).

**Author Contributions:** A.B.M.: software, formal analysis, investigation, writing—original draft, writing—review and editing, and visualization; A.A.N.: conceptualization, methodology, validation, writing—original draft and review and editing. All authors have read and agreed to the published version of the manuscript.

**Funding:** This research received no external funding.

**Institutional Review Board Statement:** Not applicable.

**Informed Consent Statement:** Not applicable.

**Acknowledgments:** The authors acknowledge the support by the Israel Science Foundation (Grant No. 843/18).

**Conflicts of Interest:** The authors declare no conflict of interest.

## Appendix A. Basic Wavevectors for the Patterns

In this paper, we considered the patterns characterized by three different sets of basic wavevectors $\mathbf{K}_i$ in the Fourier space.

For a square lattice, the basic wavevectors are perpendicular to another, Figure A1a. In the rhombic lattice, the basic wavevectors are oriented one to another by angle $\theta$. In the real space, the rhombic structures look like rectangles, Figure A1b. The most typical structures are hexagonal patterns, Figure A1c, with three basic wavevectors oriented by $2\pi/3$. That pattern is supported by the resonance interaction of disturbances forming a triad, $\mathbf{K}_1 + \mathbf{K}_2 + \mathbf{K}_3 = 0$.

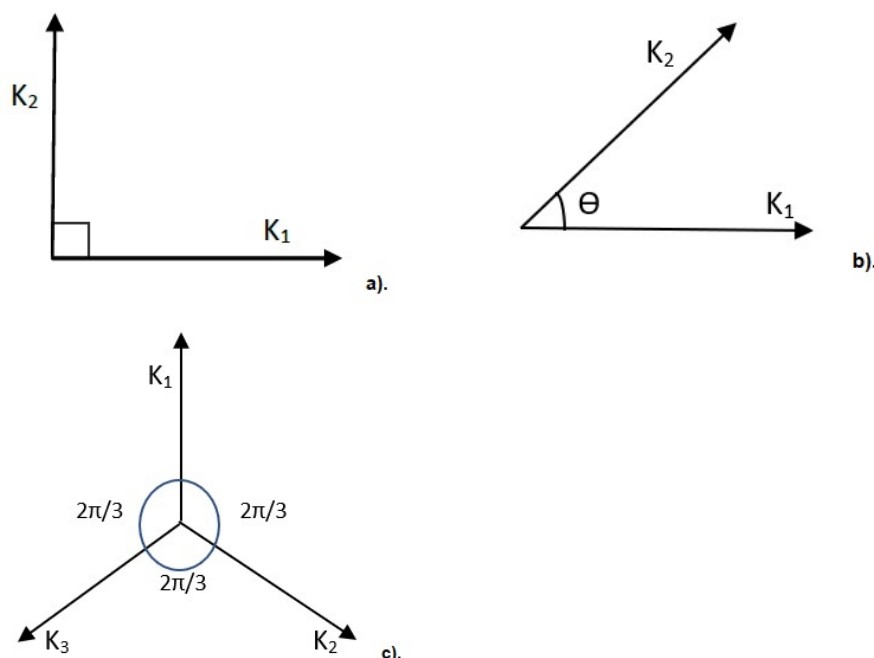

**Figure A1.** Basic wavevectors of disturbances. Panel (**a**) square/rolls lattice, (**b**) rhombic lattice, (**c**) hexagonal lattice.

**Appendix B. Coefficients of (33)–(35)**

$$a_1 = \frac{1}{3}(G + 7K^2S) - \frac{K^2M_m}{2\beta}, \qquad \tilde{a}_1 = \frac{1}{3}(G + K^2S) - \frac{K^2M_m}{2\beta},$$

$$b_1 = \frac{1}{36L}[G(12L - G) + 4K^2(G + 21L)S + 5K^4S^2] - \frac{K^2M_m}{\beta}, \qquad \text{(A1)}$$

$$\tilde{b}_1 = -\frac{(G + K^2S)(G - 12L + K^2S)}{36L} - \frac{K^2M_m}{\beta}$$

$$a_2 = \frac{G}{3}, \quad a_3 = \frac{N}{2}, \quad b_2 = \frac{G}{2}, \quad b_3 = L + N.$$

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
