# Peer review of "Marangoni Patterns in a Non-Isothermal Liquid with Deformable Interface Covered by Insoluble Surfactant"

_colloids, doi:10.3390/colloids6040053_

Round 1
Reviewer 1 Report
This paper is a review presented of the authors' recent works on the long-wave Marangoni convection of the non-isothermal liquid layer covered by insoluble surfactant. They showed the stability maps of the stationary roll with and without surfactants and concluded that the region of stable supercritical rolls shrinks in the presence of surfactant.
In the introduction, they describe that more and more experts in colloid science are interested in the thermal Marangoni related to the influence of the surfactant contamination. However, I as an expert of colloid science, don't understand how important this research is to colloid science. Since this is a review paper, it should not merely summarize the authors' previous results, but should explain how their research can be applied, citing actual experimental data.
Author Response
We added to the Introduction description of more last works in the colloid science. They are examples showing the relevance of the surfactant contamination and Marangoni stresses.
Reviewer 2 Report
This work discussed Marangoni patterns in a non-isothermal liquid with deformable interface covered by insoluble surfactant. The influence of the insoluble surfactant on the selection and modulational instability of stationary Marangoni patterns near their onset threshold were presented in detail. Particularly, the authors studied how the concentration of insoluble surfactant influences the roll stability. This is a good research paper with strong theoretical and rigorous mathematical derivation, and the data analysis is systematic and complete. Some suggestions are given as follows:
1. In this work, the elasticity parameter N was used to characterize the concentration of insoluble surfactant. What is the difference for this parameter compared to soluble surfactants? How to reflect the solubility differences of surfactants?
2. As the authors stated, the paper includes the known results published in different papers earlier, as well as new results never published yet. However, it is impossible to intuitively get the difference and significance of different research results, because the authors used a large number of descriptive texts for comparison and analysis. Thus, it is recommended that the authors analyze the results in the form of figures or tables.
3. The authors hope that the reported theory in this work can motivate more experiments on investigation of pattern formation in the system with surfactants, and will be useful for experts in colloid physics and in surfactants. However, from the perspective of a materials researchers, the new theory of this work cannot provide direct guidance and help, and its practical significance is not enough. This is mainly due to the fact that the theoretical calculation results in this paper are all in mathematical form and have not returned to the physical and chemical property parameters of materials. After the theoretical calculation is completed, these mathematical parameters should be reflected on the performance of the surfactants again.
Author Response
- In this work, the elasticity parameter N was used to characterize the concentration of insoluble surfactant. What is the difference for this parameter compared to soluble surfactants? How to reflect the solubility differences of surfactants?
Answer:
For soluble surfactants, the effects caused by surfactant-induced surface elasticity characterized by parameter N, e.g., oscillatory instability, are generally retained, but they can be slightly weakened. See, for example, Nepomnyashchy A. and Simanovskii I., Fluid Dynamics, v.23, p. 302-306 (1988) where the influence of the surfactant solubility on the Marangoni instability was studied. It should be noted that the phenomena in the case of a soluble surfactant are determined by additional parameters characterizing the adsorption/desorption kinetics and the mass transfer in the bulk (bulk diffusion coefficient, Soret coefficient etc.). See, e.g., as M. Morozov et al., Phys. Fluids, v. 25, 052107 (2013).
- As the authors stated, the paper includes the known results published in different papers earlier, as well as new results never published yet. However, it is impossible to intuitively get the difference and significance of different research results, because the authors used a large number of descriptive texts for comparison and analysis. Thus, it is recommended that the authors analyze the results in the form of figures or tables.
Answer:
The new unpublished results are presented in section 3.2 (structures on hexagonal lattice), section 3.3 (structures on rhombic lattice), and all subsections of section 5 (2D modulational instability of stationary rolls)
- The authors hope that the reported theory in this work can motivate more experiments on investigation of pattern formation in the system with surfactants, and will be useful for experts in colloid physics and in surfactants. However, from the perspective of a materials researchers, the new theory of this work cannot provide direct guidance and help, and its practical significance is not enough. This is mainly due to the fact that the theoretical calculation results in this paper are all in mathematical form and have not returned to the physical and chemical property parameters of materials. After the theoretical calculation is completed, these mathematical parameters should be reflected on the performance of the surfactants again.
Answer: We have added some data and estimates of non-dimensional parameters calculated for a real surfactant.
Reviewer 3 Report
This article discusses the patterns made by convection rolls that arise from instabilities of Marangoni flows. After a review of recent theoretical work regarding both thermocapillary and solutocapillary effects, the authors expand previous works in order to study the stability of square, hexagonal and rhombic lattices of rolls.
This article is quite technical and rather hard to read for an experimentalist in the field of colloids and interfaces. For instance, a sketch of the different patterns would be much helpful. Still, the paper is scientifically sounds and one can definitely rely on the results that are presented.
More specifically, I have the following comments:
1. The system is controlled by several dimensionless numbers (Biot, Marangoni, capillary, \ldots). But for typical experiments, some of them are very small and only a few dimensionless numbers are relevant. It would be interesting, in order to motivate new experiments, to discuss which are the regimes where we can find the interesting phenomena. Also, are these regimes realistic from the experimental viewpoint?
2. A physical understanding of the instabilities is definitely missing. Can the authors provide handwaving arguments to explain why some patterns are selected rather than others?
To conclude, I find that this paper is too specialized in its present form to attract the attention of the colloids & interface community. More explicit figures are missing, as well as a physical discussion regarding the value of the parameters and the mechanisms of the instabilities.
Author Response
This article is quite technical and rather hard to read for an experimentalist in the field of colloids and interfaces. For instance, a sketch of the different patterns would be much helpful. Still, the paper is scientifically sounds and one can definitely rely on the results that are presented.
Answer: An Appendix has been added clarifying the differences between patterns.
More specifically, I have the following comments:
1. The system is controlled by several dimensionless numbers (Biot, Marangoni, capillary, \ldots). But for typical experiments, some of them are very small and only a few dimensionless numbers are relevant. It would be interesting, in order to motivate new experiments, to discuss which are the regimes where we can find the interesting phenomena. Also, are these regimes realistic from the experimental viewpoint?
Answer: We added some estimates for non-dimensional parameters calculated for real surfactants.
- A physical understanding of the instabilities is definitely missing. Can the authors provide handwaving arguments to explain why some patterns are selected rather than others?
To conclude, I find that this paper is too specialized in its present form to attract the attention of the colloids & interface community. More explicit figures are missing, as well as a physical discussion regarding the value of the parameters and the mechanisms of the instabilities.
Answer: Concerning the physical mechanism of instability, we can say that this is essentially the thermal Marangoni instability mechanism, but its development is significantly influenced by surface deformability and Marangoni stresses caused by the advection of the surfactant along the surface.
Round 2
Reviewer 1 Report
The description of how the authors' work can be useful to experimentalists on colloidal interfaces is insufficient, but it seems to be beyond the scope of this paper, so it can be accepted as it is.
Reviewer 3 Report
The authors have proceeded to the minimal suggested modifications in order to improve the readability of the paper for the broad community interested in colloids and interfaces. In its present form, the article is now suitable for publication.